# Anatomical variations in Eurasian wolf (*Canis lupus lupus*) (Carnivora: Canidae) of the salivary glands: A histological and histochemical investigation

**Joanna Klećkowska-Nawrot**[1], **Karolina Goździewska-Harłajczuk**[1]*,
**Krzysztof Stegmann**[2], **Arkadiusz Dziech**[3], **Dariusz Łupicki**[4], **Gabriela Jędrszczyk**[5,6],
**Igor Jucenco**[5,6], **Karolina Barszcz**[7]*

1 Faculty of Veterinary Medicine, Department of Biostructure and Animal Physiology, Wrocław University of Environmental and Life Sciences, Wrocław, Poland, 2 GeoWild, Wrocław, Poland, 3 Department of Genetics, Genetics of Populations and Quantitative Traits, Wrocław University of Environmental and Life Sciences, Wrocław, Poland, 4 Museum of Natural History of the Faculty of Biology and Animal Science, Wrocław University of Environmental and Life Sciences, Wrocław, Poland, 5 Faculty of Veterinary Medicine, Second Year Student, Wrocław University of Environmental and Life Sciences, Wrocław, Poland, 6 Student Anatomical Club "Alkmeon", Wrocław, Poland, 7 Department of Morphological Sciences, Institute of Veterinary Medicine, Warsaw University of Life Sciences, Warsaw, Poland

* karolina.gozdziewska-harlajczuk@upwr.edu.pl (KG-H); karolina_barszcz@sggw.edu.pl (KB)

## Abstract

The study involved a gross anatomical description of the parotid gland, mandibular gland, monostomatic sublingual gland, polystomatic sublingual gland, and zygomatic gland in 12 adult Eurasian wolves (*Canis lupus lupus*) (wild free-ranging individuals and their zoo counterparts), including their morphometry and microscopic evaluation using hematoxylin & eosin, mucicarmine, azan trichrome, PAS, AB pH 1.0, AB pH 2.5; AB pH 2.5/PAS, and HDI stainings. Topographically, the salivary glands were located similarly to those of other terrestrial Carnivora. The parotid gland in all wolves had a shape similar to a trapezoid (four angles). The parotid duct opened onto the fourth upper premolar (P4). The parotid gland was a branched alveolar complex that produced serous secretion. In captive specimens, the mandibular gland was a complex branched tubular gland producing mucous secretion, while in free-ranging wolves it was a branched tubuloalveolar gland producing mucoserous secretion. The monostomatic sublingual gland in free-ranging wolves was a complex branched tubuloalveolar gland that produced seromucous secretion, while in captive wolves, it revealed a mucoserous secretion character. The polystomatic sublingual gland consisted of several independent packets (from 6–7 to 7–8) and was a complex branched tubuloalveolar gland with seromucous secretion. The zygomatic duct opened onto the last upper molar tooth (M3), and this gland was a complex branched tubular gland producing mucous secretion. The anatomical and histological similarities between the salivary glands of the oral cavity in captive and free-ranging wolves, compared to other terrestrial carnivores, provide valuable insights for veterinary treatments and understanding pathological conditions. These findings emphasize the need for further research on diverse populations

**Data Availability Statement:** All relevant data are within the manuscript and its Supporting Information files.

**Funding:** The APC was financed by Warsaw University of Life Sciences. The funders had no role in study design, data collection and analysis, decision to publish, or preparation of the manuscript.

**Competing interests:** The authors have declared that no competing interests exist.

of wolves and related species within the Canidae family to better understand the influence of diet on salivary gland morphology.

## Introduction

The salivary glands of the oral cavity are divided into minor salivary glands (labial, buccal, molar, palatine, lingual, zygomatic, and paracaruncular glands) and major salivary glands (monostomatic and polystomatic sublingual glands, mandibular gland, and parotid gland) [1]. Minor salivary glands, as small packets, are present in the mucous and submucous membranes of the oral cavity [2,3]. The major salivary glands provide saliva to the oral vestibule and the oral cavity proper through a simple excretory duct [1–3]. The zygomatic gland is present exclusively in the Carnivora and is derived from the dorsal buccal glands located in the zygomatic region [1]. This gland is large in size and has one larger duct opening at the zygomatic papilla and 2 to 4 smaller ducts opening into the buccal vestibule. Therefore, it was previously named the orbital salivary gland [1].

The composition of saliva in mammals depends on various factors, including the age and sex of the individual, but also its physical health, environmental conditions (time of day and year), physical activity, diet, and the source of stimuli leading to saliva formation (i.e. mechanical, chemical, or psychoneurological) [4]. Differentiation in composition is also visible between different individuals of the same species [5–7]. These factors, in addition to influencing saliva composition, have an impact on the size of the oral cavity glands, their architecture, and the nature of the secretion produced (serous, mucous, or mixed–mucoserous or seromucous) [5–8].

Morphology of the salivary glands of the oral cavity of species in the Canidae family was described in detail for the domestic dog (*Canis lupus familiaris*) [2,3,9–25]. Furthermore, anatomical and histological studies of major salivary glands were described for the red fox (*Vulpes vulpes*) [26], crab-eating fox (*Cerdocyon thous*) [27,28], maned wolf (*Chrysocyon brachyurus*) [29], pampas fox (*Lycalopex gymnocercus*) [28], fennec fox (*Vulpes zerda*) and South African painted dog (*Lycaon pictus pictus*) [30]. The literature describing salivary gland pathologies in the Canidae family is limited to the domestic dog [31–35], red fox and coyote (*Canis latrans*) [36].

The main aim of our research was to determine whether there are differences in the morphology of salivary glands in the same species living in different environments (free-ranging and captive Eurasian wolves) which consequently have different diets. This study also provides a basis for a better understanding of salivary gland disorders in the general field of veterinary medicine. Our results may be helpful in diagnosing diseases of these anatomical structures, especially in wild representatives of the Canidae family found in zoos or national parks. We compared the results of our research with other representatives of the Canidae family due to their diverse diet, which enriches our analysis.

## Materials and methods

### Collection of specimens

The salivary glands examined were obtained from 12 adult (3 males and 9 females) Eurasian wolves (*Canis lupus lupus*) (hereafter referred to as "wolves") (Fig 1A). Two female wolves were acquired from Wroclaw Zoological Garden (Poland) (weighing 30–34 kg), while ten free-ranging wolves (3 males weighing 38–42 kg and 7 females weighing 28–36 kg) were sourced

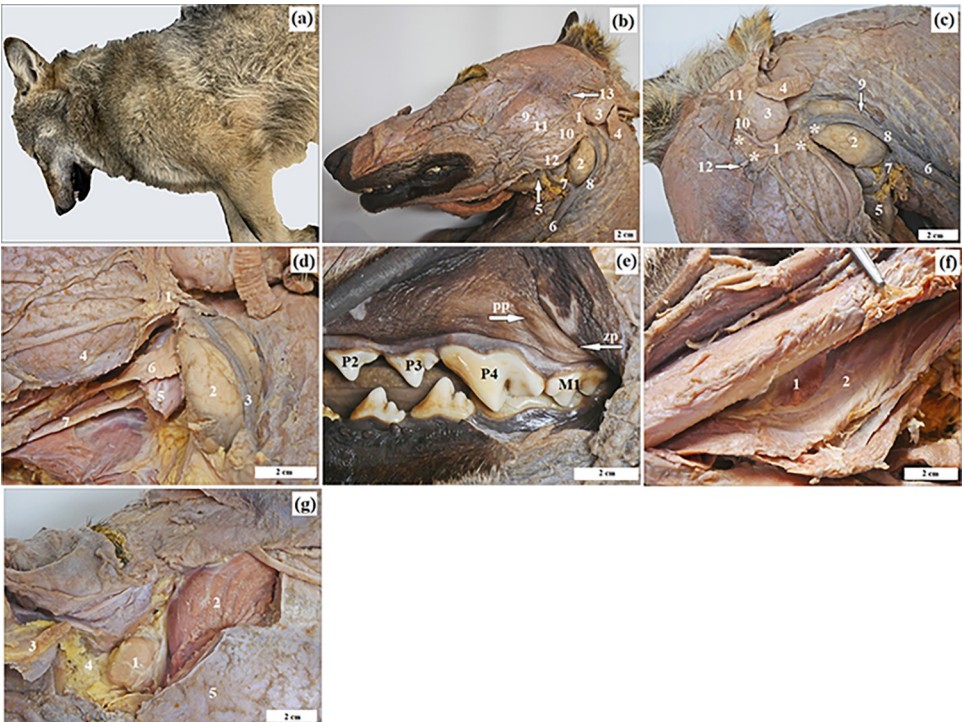

**Fig 1. The macrograph of the major salivary glands and zygomatic gland in free-ranging and captive Eurasian wolves.** a) free-ranging wolf; b) 1 –parotid gland, 2 –mandibular gland, 3 –auricular cartilage, 4 –zygomaticoauricularis muscle, 5 –mandibular lymph nodes, 6 –external jugular vein, 7 –linguofacial vein, 8 –maxillary vein, 9 –masseter muscle, 10 –parotid duct, 11 and 12 –buccolabial branches (facial nerve); c) 1 –parotid gland (white asterisk), 2 –mandibular gland, 3 –auricular cartilage, 4 –zygomaticoauricularis muscle, 5 –mandibular lymph nodes, 6 –external jugular vein, 7 –linguofacial vein, 8 –maxillary vein, 9 –caudal auricular vein, 10 –superficial temporal vein, 11 –zygomatic muscle, 12 –auriculopalpebral nerve, red asterisk–four angles of the parotid gland; d) 1 –parotid gland, 2 –mandibular gland, 3 –maxillary vein, 4 –masseter muscle, 5 –digastric muscle, 6 –monostomatic sublingual gland, 7 –hypoglossal nerve; e) pp (white arrow)—position of the parotid papilla, zp (white arrow)–position of the zygomatic papilla, P2-P3-P4 –premolar teeth, M1 –molar tooth; f) 1 –polystomatic sublingual gland, 2 –major sublingual duct,3– digastric muscle; g) 1 –zygomatic gland, 2 –temporal muscle, 3 –zygomatic arch, 4 –extraperiorbital fat body, 5 – masseter muscle. Bar = 2 cm.

from natural conditions. Wolves from natural conditions died in car accidents in Lower Silesia, Kuyavian–Pomeranian and Greater Poland voivodeships (Poland). Upon notification of such events, carcasses were promptly transported by one of the authors, foresters, or a veterinary inspector to the university freezers to prevent decomposition. The captive wolves were obtained following their natural deaths, and no animals were killed for the purpose of this study. The entire research material was collected from 2017 to 2024 in the Division of Animal Anatomy and Museum of Natural History at Wrocław University of Environmental and Life Sciences.

## Ethical statement

Personal permits (for *post-mortem* collection from Wrocław Zoological Garden) were obtained from the Country Veterinary Officer in Wrocław (Poland) (No. PIW Wroc. UT-45/ 5/16 –Dr. Joanna Klećkowska-Nawrot, No. PIW Wroc. UT- 45/6/16 –Dr. Karolina Goźd- ziewska-Harłajczuk). The acquisition and storage of free-ranging Eurasian wolf carcasses for Wrocław University of Environmental and Life Sciences was carried out with permission from the General Directorate for Environmental Protection in Warsaw (Poland) (No.

DZP-WG.6401.179.2022.BP). According to Polish and European law, studies on tissues obtained *post-mortem* do not require the approval of the Ethics Committee (Journal of Laws of the Republic of Poland, the Act of 15[th] January 2015, on the protection of animals used for scientific or educational purposes (Directive 2010/63/EU of the European Parliament and of the Council of 22[nd] September 2010 on the protection of animals used for scientific purposes).

### Morphometry

The measurements of the examined glands were performed on fresh material before preservation in formalin for macroscopic analysis (preparation) and microscopic evaluation. The morphometric measurements (length, width, thickness) of examined glands (left and right side) were conducted using a digital caliper with a resolution of 0.01 mm and an accuracy of +/- 0.02mm (>100mm) (Handy Worth, Poland). Anatomical descriptions of the major salivary glands and zygomatic gland were based on *NAV* [1].

### Gross anatomy

The preservation of each wild specimen occurred within 5 hours of reporting the discovery of the wolf carcass. The animals in the zoo were subjected to necropsy immediately after death, and then transferred to the cold storage. The morphology of the major salivary glands and zygomatic gland was described using methods in topographic anatomy (holotopy and syntopy) [37]. Pictures were captured with a Nikon D300s camera equipped with a Tamron AF 17–50 mm F/2.8 [IF] ∅ 67 lens.

### Histological and histochemical examination

All analyzed oral glands were placed in 4% buffered formaldehyde for at least 72 hours and then dehydrated using a 75%, 96% and 100% ethanol solution in the ETP vacuum tissue processor (RVG3, Intelsint, Italy). Paraffin blocks were cut in Micron HM310 microtome into 5μm sections. The following stains were performed: hematoxylin and eosin, azan trichrome and mucicarmine as previously described by Burck [38]. For the histological examination of salivary glands periodic acid-Schiff (PAS); alcian blue pH 1.0 (AB pH 1.0); alcian blue pH 2.5 (AB pH 2.5); alcian blue pH 2.5 PAS (AB pH2.5/PAS) and Hale's dialyzed iron (HDI) were performed to evaluate the composition of glandular secretion [39]. The histochemical evaluation of examined oral glands was described in accordance with the method provided by Spicer and Henson, where (-) indicated a negative reaction; (-/+) and (+) denoted a weak reaction; (++) signified a mild reaction; and (+++) represented a strong reaction [39]. The slides were then analyzed using the Zeiss Axio Scope A1 light microscope (Carl Zeiss, Jena, Germany). Histological descriptions of examined salivary glands of the oral cavity structure were based on *NAH* [40].

## Results

### The salivary glands of the oral cavity morphometry

The largest salivary glands in examined free-ranging and captive wolves were the mandibular glands, followed by the parotid, sublingual (polystomatic and then monostomatic), and finally the zygomatic. The size of the parotid gland was consistently reduced in all wolves studied. The mandibular gland was largest in male free-ranging wolves and smallest in female free-ranging wolves. The zygomatic gland in male free-ranging wolves was the longest, while in terms of width and thickness, it was very similar in all wolves examined (Tables 1 and S1). The longest excretory duct was associated with the major sublingual gland, followed by mandibular and parotid glands. The measurements indicated that the parotid duct was longer in male free-

Table 1. Morphometry (mm) of the major salivary glands, zygomatic gland and ducts in the captive and free-ranging Eurasian wolves (*Canis lupus lupus*). Mean ±S.D.

| Measurement | PAROTID GLAND | | | PAROTID DUCT | MANDIBULAR GLAND | | | MANDIBULAR DUCT | MONOSTOMATIC SUBLINGUAL GLAND | | | MAJOR SUBLINGUAL DUCT | POLYSTOMATIC SUBLINGUAL GLAND | | | ZYGOMATIC GLAND | | |
|---|---|---|---|---|---|---|---|---|---|---|---|---|---|---|---|---|---|---|
| | length | width | thickness | length | length | width | thickness | length | length | width | thickness | length | length | width | thickness | length | width | thickness |
| FEMALE CAPTIVE WOLVES | 34.462 ±0.9 | 7.385 ±1.07 | 4.997 ±0.2 | 81.41 ±1.6 | 42.005 ±1.1 | 21.945 ±0.5 | 12.492 ±1.2 | 109.255 ±3.4 | 18.52 ±0.1 | 11.1 ±0.2 | 5.6 ±0.2 | 104.53 ±1.4 | 28.73 ±0.8 | 8.877 ±0.5 | 3.385 ±0.2 | 20.79 ±0.6 | 20.257 ±0.6 | 11.96 ±0.2 |
| MALE WILD FREE-RANGING WOLF | 35.941 ±1.7 | 7.556 ±0.7 | 6.141 ±0.3 | 82.835 ±3.2 | 50.941 ±1.5 | 25.315 ±1.01 | 16.638 ±0.4 | 118.5 ±1.1 | 19.336 ±1.8 | 12.593 ±0.5 | 5.855 ±0.2 | 111.195 ±2.4 | 31.131 ±0.7 | 10.535 ±0.4 | 4.05 ±0.07 | 24.076 ±0.6 | 20.336 ±0.6 | 12.345 ±0.2 |
| FEMALE WILD FREE-RANGING WOLF | 33.206 ±1.7 | 7.247 ±0.4 | 5.867 ±0.2 | 79.446 ±1.2 | 38.342 ±3.5 | 20.908 ±1.6 | 13.822 ±0.7 | 108.963 ±2.6 | 18.212 ±0.5 | 11.399 ±0.6 | 5.727 ±0.3 | 103.815 ±1.9 | 29.649 ±0.9 | 8.23 ±0.6 | 4.035 ±0.1 | 21.416 ±0.8 | 19.687 ±0.6 | 11.566 ±0.7 |

ranging wolves. The mandibular duct was longest in male free-ranging wolves, while the difference in duct length between female captive wolves and female free-ranging wolves was very small. The major sublingual duct was the longest in male free-ranging wolves, while in examined females the length of the duct was very similar (Table 1).

## The parotid gland

**Gross anatomy and histology.** The parotid gland in all examined wolves was situated between the ramus of the mandible and the transverse process of the atlas, within the retromandibular fossa at the base of the auricular cartilage. Externally, it was enveloped by the parotid fascia and a well-developed parotidoauricular muscle. The parotid gland resembled a trapezoid shape. (Fig 1B and 1C). It possessed four angles: rostralodorsal (wide), situated in the preauricular part bordering the masseter muscle; caudalodorsal (narrow), positioned below the auricular cartilage bordering the zygomaticoauricular muscle; rostraloventral (wide), adjacent to the preauricular part and the parotid duct; and the largest angle, directed caudaloventrally, which came into contact with a small region of the mandibular gland (Fig 1C). The parotid duct exited the gland at the lower end of its anterior edge, traversed along the surface of the masseter muscle and then pierced through the cheek at the level of the fourth premolar teeth (P4) (sectorial teeth), ultimately emptying into the oral vestibule at the parotid papilla (Fig 1B and 1E). This gland, in both free-ranging and captive wolves, was encased in a thick connective capsule composed of dense connective tissue housing numerous intralobular ducts, blood vessels and nerves. The stroma of the gland in all wolves was divided into large lobes by thick septa, further subdivided into smaller segments (Fig 2A).

These lobes exhibited various shapes such as oval, elongated, triangular, or quadrangular (Fig 2A). The intralobular ducts were characterized by taller columnar cells and narrow lumina (Fig 2B). The striated ducts composed of cuboidal or columnar cells containing basal striations (Fig 2C and 2D). The parotid gland in both free-ranging and captive wolves was identified as a compound alveolar gland producing serous secretion (Figs 2C, 2D and 3A–3E and Table 2). In two captive wolves, a noteworthy presence of lymphocytes surrounding striated ducts was noted (Fig 2E). Mucicarmine staining revealed a negative reaction in the serous acini of all studied wolves (Fig 2F).

## The mandibular gland

**Gross anatomy and histology.** The mandibular gland was located in the retromandibular fossa, bounded anteriorly by the posterior edge of the ramus of the mandible, superiorly and posteriorly by the transverse process of the atlas and inferiorly by the basihyoid body (Fig 1C). This gland was covered with a thick fibrous capsule. The dorsal part of this gland was covered by the parotid gland, while the remaining ventral part lay superficially under the skin. Moreover, this salivary gland was in contact with the masseter muscle, linguofacial vein, maxillary vein, caudal auricular vein and mandibular lymph nodes (Fig 1A and 1B). It was oval in shape with a slightly concave rostral margin (Fig 1A and 1B). The mandibular duct ran between the mylohyoid muscle and hyoglossus muscle and opened into the oral cavity proper at the sublingual caruncle (Fig 1E).

The mandibular gland had a thick connective capsule, which sent thick and thin interlobar septa dividing the stroma into dominant large lobes and a few medium-sized and small lobes (Figs 4A and 5A). The duct system (intralobular ducts and striated ducts) was well-developed (Figs 4B, 4C and 5B–5D).

In free-ranging wolves this gland was a branched tubuloalveolar gland that produced mucoserous secretion (Figs 4B, 6A, 6C, 6Ee 6G and 6I and Table 2), while in captive wolves it was a

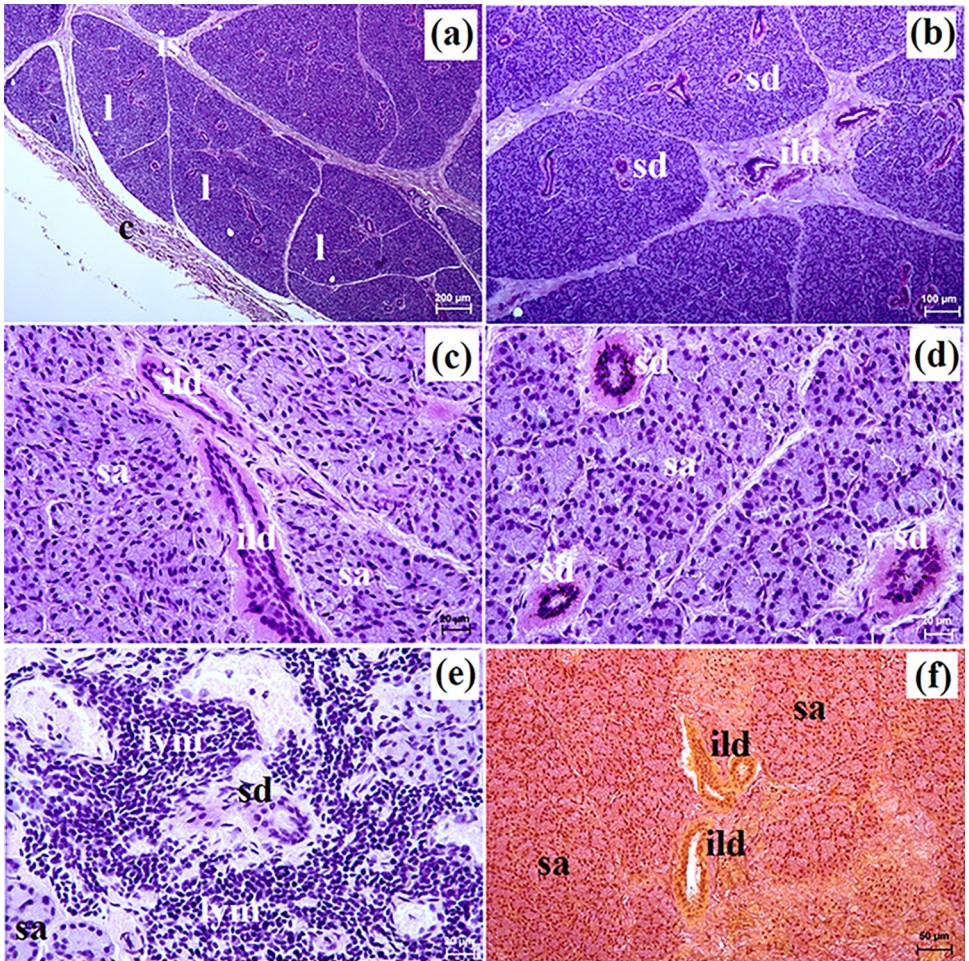

**Fig 2. The histological images of the parotid gland in free-ranging and captive Eurasian wolves.** c–capsule, ild–intralobular duct, is–interlobar septa, l–lobes, lym–lymphocytes (captive wolves), sa–serous acini, sd–striated duct. a–e = H&E stain; f = mucicarmine stain. Bar = a– 100 μm; c, d, e– 20 μm; f– 50 μm.

complex branched tubular gland producing mucous-type secretion (Figs 5C, 5D, 6B, 6D, 6F, 6H and 6J and Table 2). In one male and three female free-ranging wolves, lymphocytes were (Fig 4D). In captive wolves, mucicarmine staining showed a slight (+) reaction in the mucous cells (Fig 5E and 5F) and in free-ranging wolves, a slight (+) reaction in the tubules and a negative reaction in the serous demilunes were observed (Fig 4E and 4F).

## The monostomatic sublingual gland

**Gross anatomy and histology.** The monostomatic sublingual gland was situated under the mucosa of the lateral sublingual recess of the floor of the mouth. It was closely adjacent to the mandibular gland, from which it was separated by a septum of a common connective tissue capsule (Fig 1D). The shape of this salivary gland was oval with a slightly marked indentation on the rostral margin. The major sublingual duct ran together with the mandibular duct and entered the oral cavity proper at the sublingual caruncle (Fig 1F).

The monostomatic sublingual gland was surrounded by a thick connective tissue capsule, which, sending a thick interlobar septa, divided this gland into large lobes, which were then

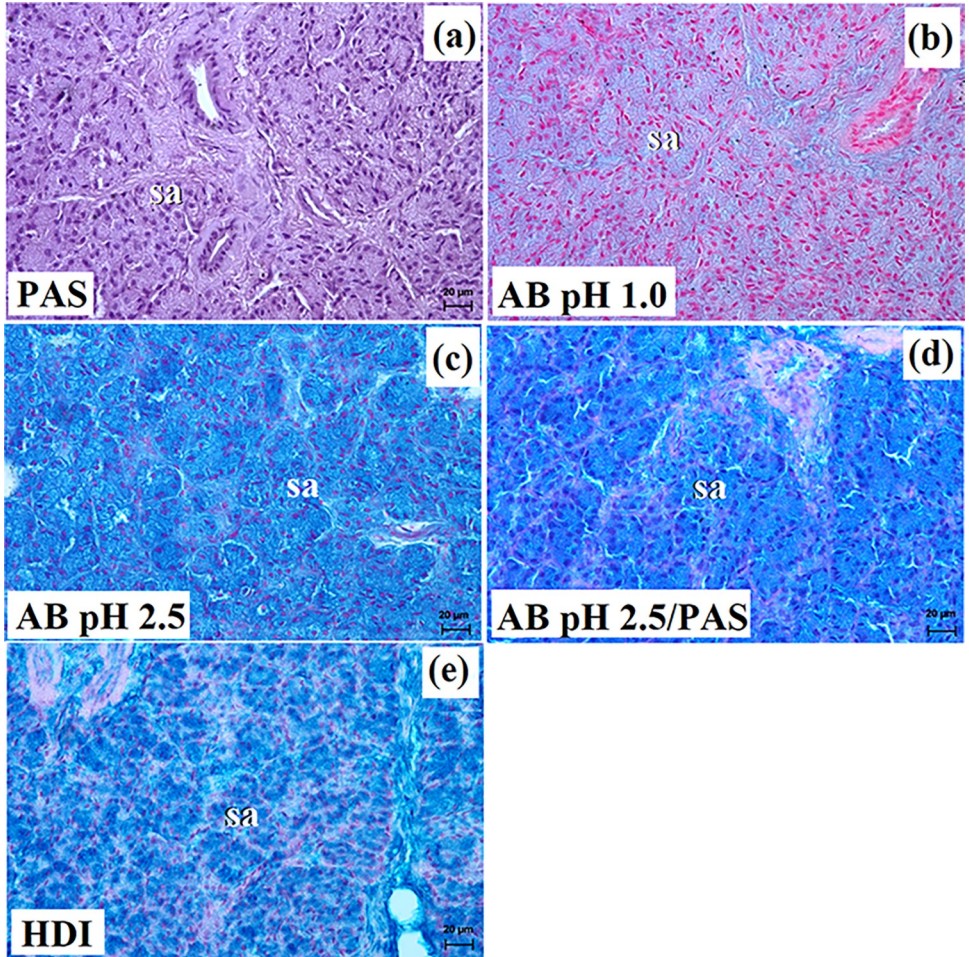

**Fig 3. The histochemical images of the parotid gland in captive and free-ranging Eurasian wolves.** sa–serous acini.
Bar = 20 μm.

divided into smaller segments with different shapes (oval, elongated, triangular, quadrangular) (Figs 7A and 8A).

In free-ranging wolves, numerous adipose cells were observed, both in the interlobar septa and between the secretory units (Fig 8A). The intralobular ducts in free-ranging wolves were oval in cross section with a wide lumen (Fig 8C), while in captive wolves, they were most often elongated with a narrow lumen (Fig 7B). The monostomatic sublingual gland in free-ranging wolves was a complex branched tubuloalveolar gland producing seromucous secretion, where there was a clear predominance of serous cells (Figs 8B, 8C, 9A, 9C, 9E, 9G and 9I and Table 2), while in captive wolves, it was a complex branched tubuloalveolar gland with a predominance of tubules with mucoserous character of secretion (Figs 7D, 9B, 9D, 9F, 9H and 9J and Table 2). Furthermore, in a male and three female free-ranging wolves, the presence of numerous lymphocytes was observed (Fig 9D). Mucicarmine staining showed a negative reaction in the serous demilunes and a medium (++) reaction in the mucous tubules and ducts in all examined wolves (Figs 7E, 7F, 8E and 8F).

**Table 2. Comparative histochemical characterization of the examined major salivary glands and zygomatic gland in captive and free-ranging Eurasian wolves (*Canis lupus lupus*).**

| HISTOCHEMICAL STAIN | PAROTID GLAND | MANDIBULAR GLAND | | MONOSTOMATIC SUBLINGUAL GLAND | | POLYSTOMATIC SUBLINGUAL GLAND | ZYGOMATIC GLAND | |
|---|---|---|---|---|---|---|---|---|
| | captive and free-ranging wolves | captive wolves | free-ranging wolves | captive wolves | free-ranging wolves | captive and free-ranging wolves | captive wolves | free-ranging wolves |
| PAS | (−) | (−) | (+++) serous demilunes and (++) tubules | (−) | (−) serous demilunes and (++) tubules | (−) | (+++) | (−) |
| AB pH 1.0 | (−) | (+++) | (−) serous demilunes and (+++) tubules | (−) serous demilunes and (+++) tubules | (−) serous demilunes and (++) tubules | (−) serous demilunes and (+++) tubules | (+++) | |
| AB pH 2.5 | (+++) | (++) | (+++) serous demilunes and (++) tubules | (+++) serous demilunes and (++) tubules | (++) serous demilunes and (+++) tubules | (+) serous demilunes and (+++) tubules | (+++) | |
| AB pH 2.5/PAS | (+++) (blue color) | (+++) (blue color) | (+) serous demilunes (magenta color) and (++) tubules (blue color) | (++) serous demilunes (magenta color) and (+++) tubules (blue color) | (+) serous demilunes (magenta color) and (+++) tubules (blue color) | (++) serous demilunes (magenta color) and (+++) tubules (blue color) | (+++) (magenta color) | (+++) (blue color) |
| HDI | (+++) | (+) | (−) serous demilunes and (++) tubules | (−) serous demilunes and (+) tubules | (−) serous demilunes and (+++) tubules | (−) serous demilunes and (+++) tubules | (+++) | |

## The polystomatic sublingual gland

**Gross anatomy and histology.** The polystomatic sublingual gland was located between the mucosa of the oral cavity proper and the mylohyoid muscle and styloglossus muscle (Fig 1F). It was narrow and long. It consisted of several (6–7 to 7–8) independent packets, each of which had its own leading duct, a minor sublingual duct opening directly into the floor of the oral cavity proper. The topographic range of the monostomatic and polystomatic sublingual gland extends from the angle of the mandible to the palatoglossal arch.

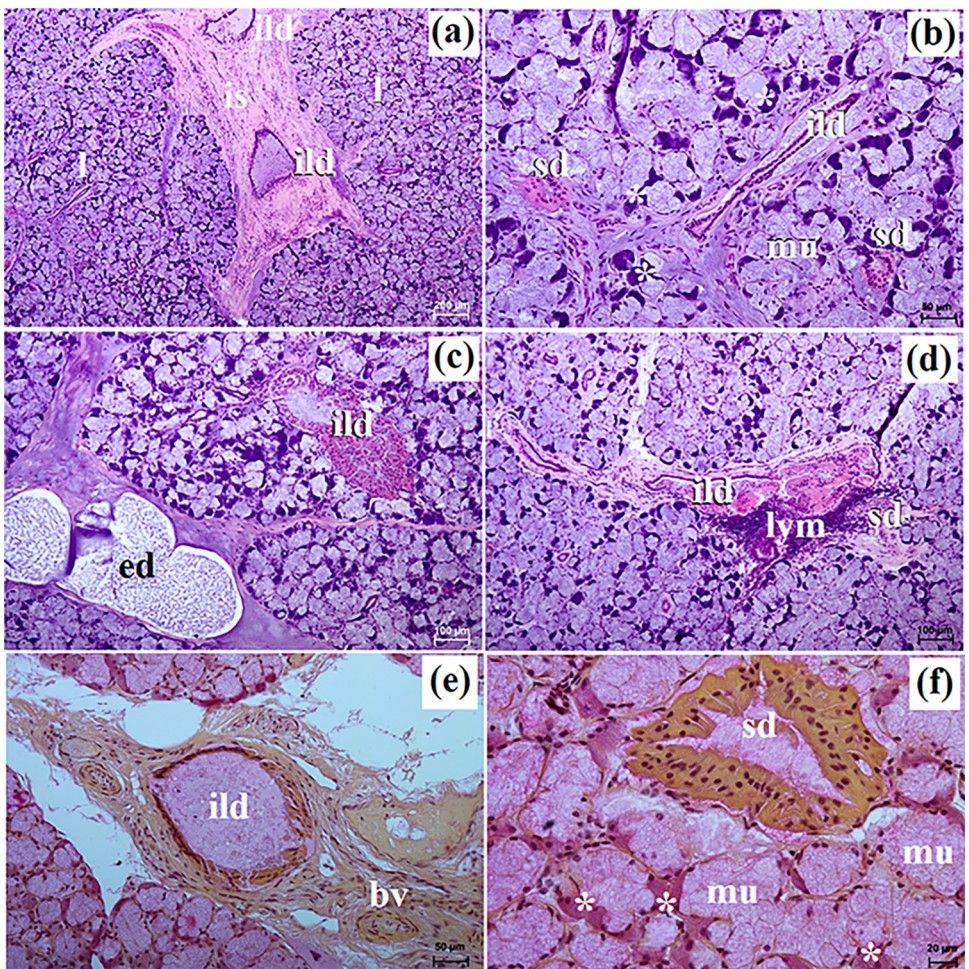

**Fig 4. The histological images of the mandibular gland in free-ranging Eurasian wolves.** bv–blood vessels, ed–excretory ducts, ild–intralobar duct, is–interlobular septa, l–lobes, lym—lymphocytes, mu–mucous units, sd–striated duct, white asterisk–serous demilunes. a–d = H&E stain; e–f = mucicarmine stain. Bar = a– 200 μm; c, d– 100 μm; b, e–50 μm; f– 20 μm.

In all examined wolves the polystomatic sublingual gland was covered by a thick connective tissue capsule and was composed of numerous very large lobes and less numerous smaller lobes of various shapes (oval, elongated, triangular, quadrangular) formed by thin and thick intralobar septa without secondary division (Fig 10A). Few adipose cells were observed between secretory units (Fig 10A and 10B). The duct system (intralobular ducts and striated ducts) was well-developed (Fig 10B and 10C). In both free-ranging and captive wolves, it was a complex branched tubuloalveolar gland with seromucous secretion (Figs 10D and 11A–11E and Table 2). Furthermore, in the two captive female wolves, numerous clusters of lymphocytes were visible (Fig 10E). Mucicarmine staining showed in all wolves a negative reaction in the serous demilunes and a slight (+) reaction in the mucous tubules (Fig 10F).

## The zygomatic gland

**Gross anatomy and histology.** The zygomatic gland was observed in the anterior part of the pterygopalatine fossa, surrounded by extraperiorbital fat body (Fig 1G). It was oval and adjacent to the side zygomatic arch, masseter muscle and temporal muscle, while from the

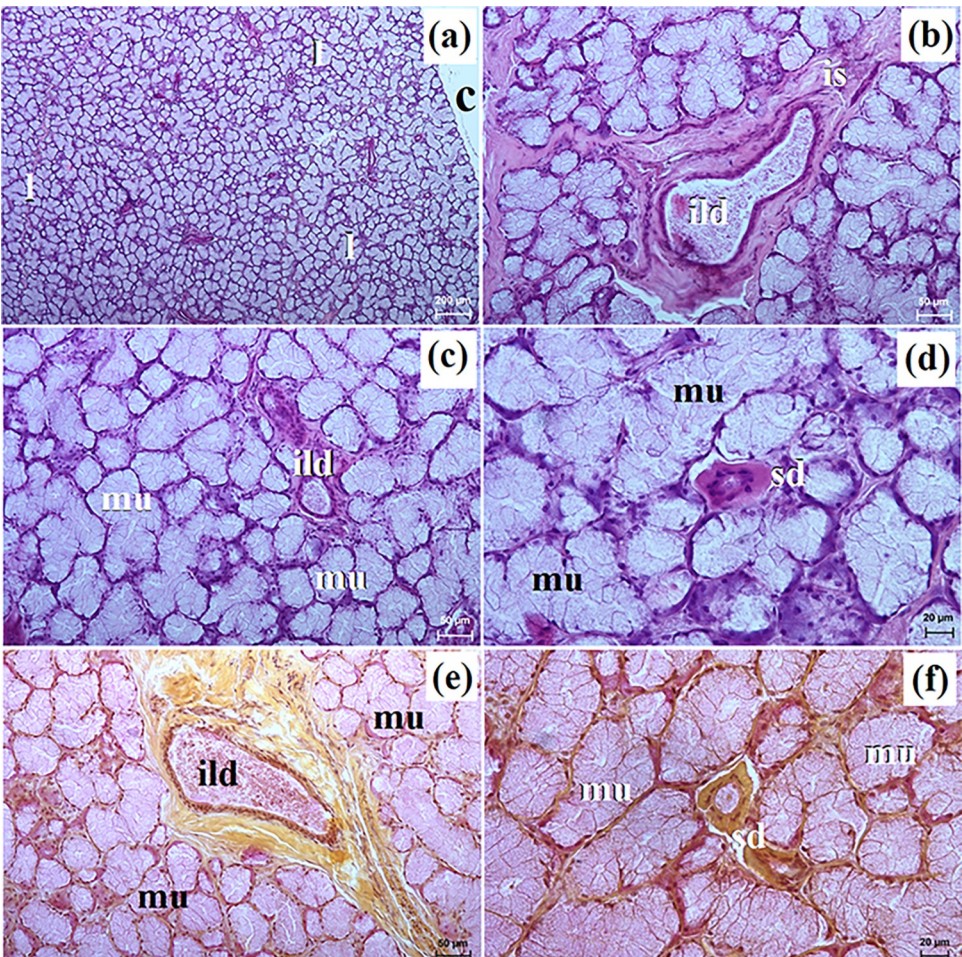

**Fig 5. The histological images of the mandibular gland in captive Eurasian wolves.** c–capsule, ild–intralobular duct, is–interlobar septa, l–lobes, mu–mucous units, sd–striated duct. a–d = H&E stain; e–f = mucicarmine stain. Bar = a– 200 μm; b, c, e– 50 μm; d, f– 20 μm.

medial side to the periorbita, pterygoid muscle, maxillary artery and maxillary nerve. It formed an irregular, slightly rounded cluster of small packets surrounded by a thick fibrous capsule. The zygomatic gland had 4 to 5 ducts (1 large major duct of the zygomatic gland and 3 to 4 small ducts called minor ducts of the zygomatic gland) that opened into the oral vestibule at the level of the last upper molar (M3) (Fig 1E).

The zygomatic gland was surrounded by a thick connective tissue capsule, within which numerous clusters of adipose cells were found (Figs 12A and 13A).

In free-ranging wolves, thin intralobar septa divided the gland into large but poorly defined lobes (Fig 12A) whereas in captive wolves, both thin and thick septa were present, which divided the gland into numerous large lobes and occasional small lobes (Fig 13A). Furthermore, in captive wolves, it was observed that the intralobular ducts were surrounded by abundant connective tissue (Fig 13B). The zygomatic gland in free-ranging and captive wolves was a complex branched tubular gland that produces mucous secretion (Figs 12C, 13D and 14A–14J and Table 2). Mucicarmine stain revealed a strong (+++) reaction in mucous cells in captive wolves (Fig 14E and 14F), while for free-ranging wolves, a medium (++) reaction was observed (Fig 13E and 13F).

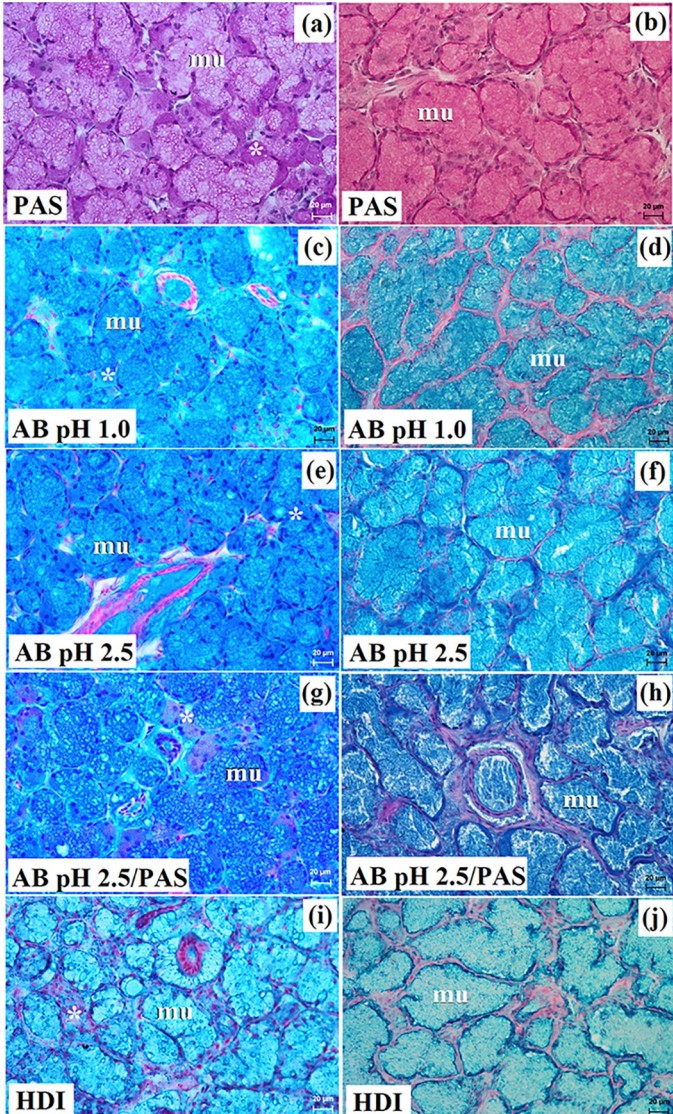

**Fig 6.** The histochemical images of the mandibular gland in free-ranging Eurasian wolves (a, c, e, g, i) and captive wolves (b d f, h, j). mu–mucous units, white asterisk–serous demilunes. Bar = 20 μm.

## Discussion

The wolves are considered generalist carnivores, and their diet is highly diversified, depending on environmental conditions and the species composition of the ecosystems in their territories [41]. They can feed on wild ungulates, livestock, small vertebrates, carcasses, plants and anthropogenic leftovers [42]. An important factor influencing their diet composition is the availability and density of ungulates in the area [43]. Other factors influencing their diet include prey species composition, their condition, age-sex structure, defense strategy, pack size, hunting habits and environmental conditions (season, weather, etc.) [44]. In Poland, similar to many other regions of their distribution, the most abundant prey species in the wolves' diet are red deer (*Cervus elaphus*), European roe (*Capreolus capreolus*) and to a lesser extent, wild boar (*Sus scrofa*). Additionally, wolves prey on fallow deer (*Dama dama*), European hare (*Lepus europaeus*), Eurasian beaver (*Castor fiber*) and smaller animals such as raccoon dogs

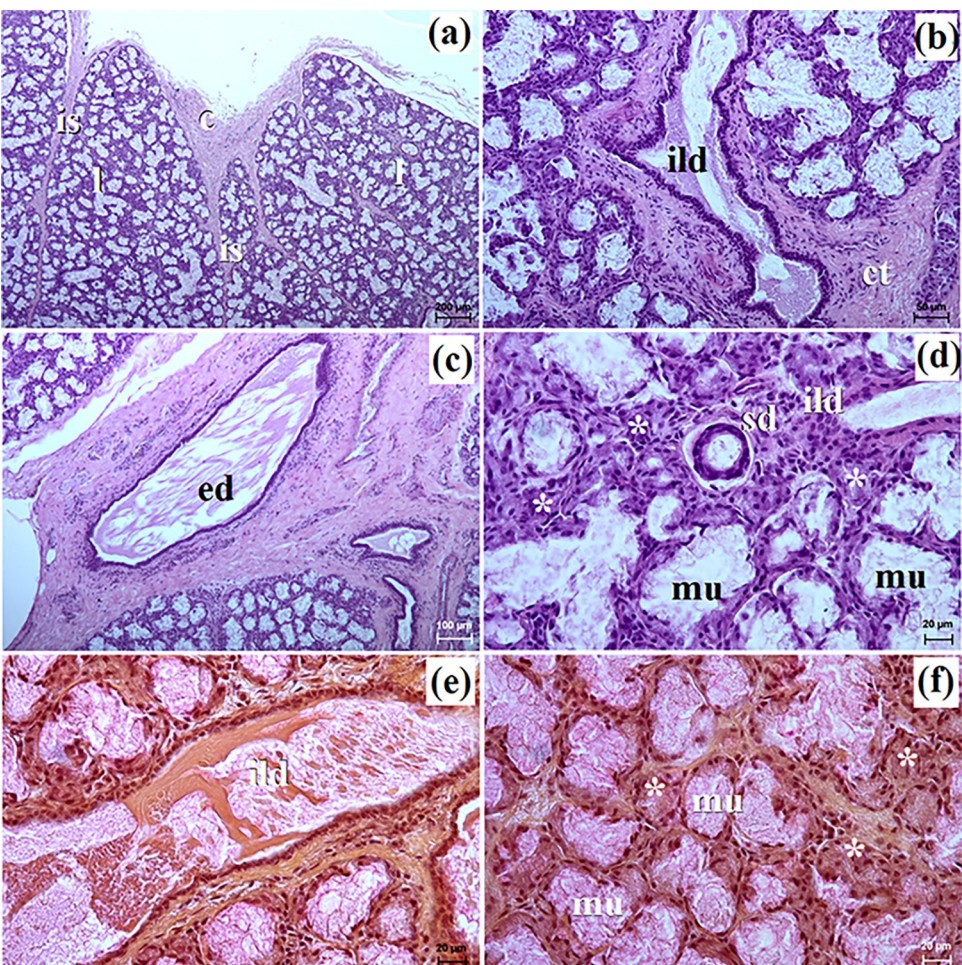

**Fig 7. The histological images of the monostomatic sublingual gland in captive Eurasian wolves.** c–capsule, ct–connective tissue, ed–excretory duct, ild–intralobular duct, is–interlobar septa, l–lobes, mu–mucous units, sd–striated duct, white asterisk–serous demilunes. a–d = H&E stain; e–f = mucicarmine stain. Bar = a– 200 μm; b– 50 μm; c– 100 μm; d, e, f– 20 μm.

(*Nyctereutes procyonoides*), red foxes, rodents, birds, amphibians, reptiles and insects [41,45,46]. Moreover, free-ranging wolves in Poland also hunt livestock, mainly cattle and sheep [47–49]. Some attacks on domestic dogs (*Canis lupus familiaris*) were recorded, but they are thought to result from territorial competition between these closely related species, with dogs perceived as intruders or rivals in wolf territory [48,49].

In zoos, it is not feasible to provide an identical diet to that found in nature. Predator animals are primarily fed beef, with some diets supplemented with veal, beef and fish. Wolves may also be given vegetables or fruits, such as pumpkins and watermelons. Additionally, their diet is often supplemented with a mixture of specially selected vitamins and minerals [50]. The diet of wolves at the Wroclaw Zoological Garden primarily consists of beef with bones, supplemented with poultry meat (chicken, turkey and quail), offal, fish (mackerel, herring), rabbits and dietary supplements over the years (information from Wroclaw Zoological Garden).

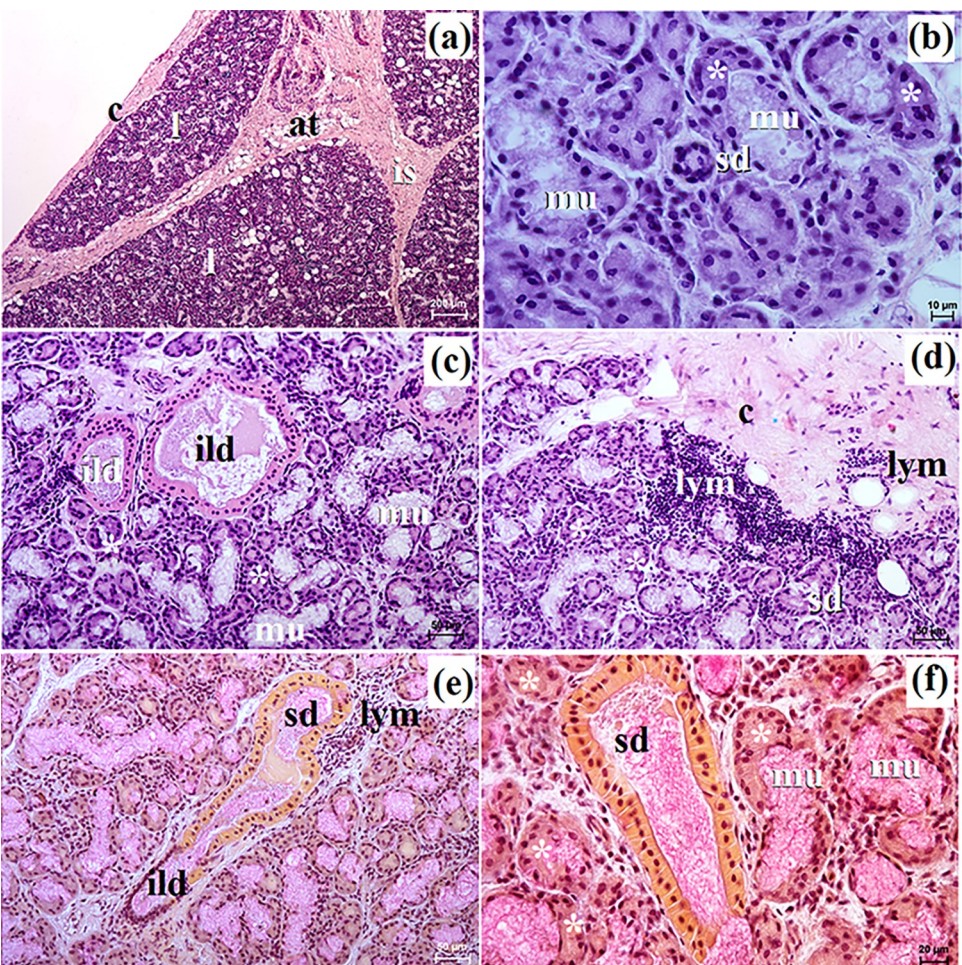

**Fig 8. The histological images of the monostomatic sublingual gland in free-ranging Eurasian wolves.** at–adipose tissue, c–capsule, ild–intralobular duct, is–interlobar septa, l–lobes, lym–lymphocytes, mu–mucous units, sd–striated duct, white asterisk–serous demilunes. a–d = H&E stain; e–f = mucicarmine stain. Bar = a– 200 μm; b– 10 μm; c, d, e– 50 μm; f– 20 μm.

## The parotid gland

The parotid gland localization in all examined wolves was similar to other Canidae such as the domestic dog [10,11,23,24], Baladi dog [26], crab-eating fox [27,28], red fox [28], fennec fox [30], South African painted dog [30] and pampas fox [28].

Morphometric studies of the parotid gland in all examined wolves showed, that this gland was proportionally similar in size to the domestic dog (weighing 10–15 kg) but smaller in red fox (weighing 4–7 kg) and South African painted dog [10,26,30]. Because of the variations in body size between breed and mongrel dogs, as well as differences in size among other Canidae species, the size of all examined salivary glands will differ among these animals. However, in the Baladi dog, this gland was almost twice as long as in the examined wolves but much narrower [26]. According to Krysiak and Świeżyński [51], the parotid gland in the domestic dog is smaller than the mandibular salivary gland. However, according to the research of Pereira and Faria Junior [27] in the crab-eating fox the parotid gland is larger than the mandibular salivary gland.

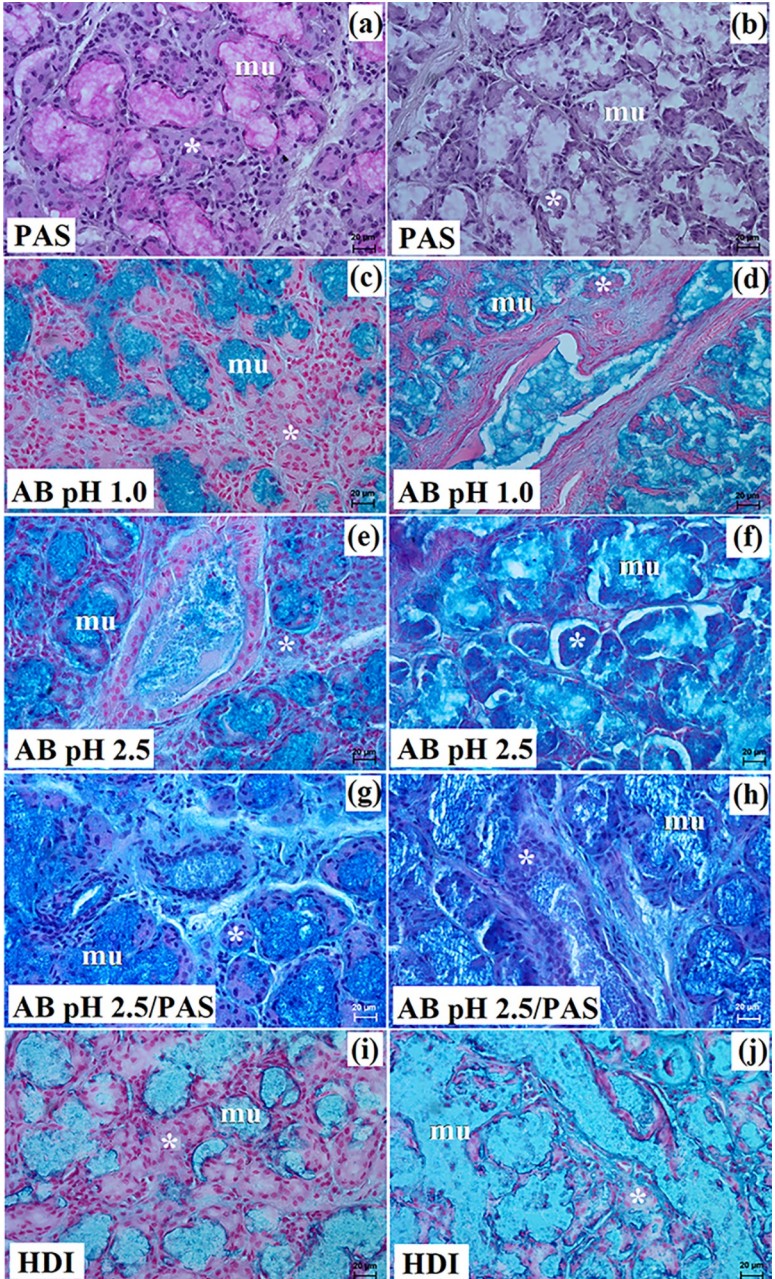

**Fig 9.** The histochemical images of the monostomatic sublingual gland in free-ranging Eurasian wolves (a, c, e, g, i) and captive wolves (b, d, f, h, j). mu–mucous units, white asterisk–serous demilunes. Bar = 20 μm.

The shape of the parotid gland in wolves studied was close to a trapezoid and had a clearly marked four angles. Whereas in the domestic dog, the Baladi dog, the fennec fox, the red fox and the crab-eating fox this salivary gland was triangular in shape [10,23,24,26,27,30,51,52], and in the South African painted dog had a clearly marked five angles [30].

The parotid duct in the examined wolves pierced the cheek wall at the level of the fourth upper premolars (P4), similarly to the Baladi dog [26], between the third and fourth upper premolar tooth in the red fox and domestic dog [3,26], at the level of the third premolar tooth in

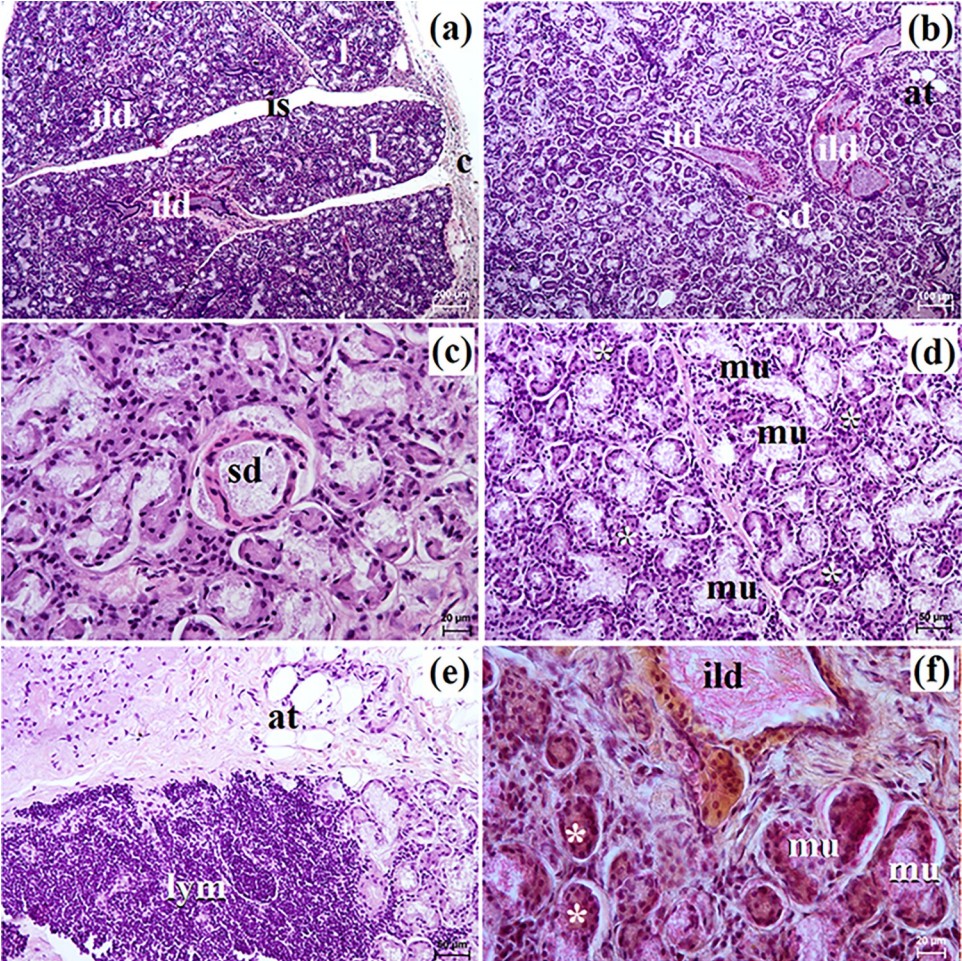

**Fig 10. The histological images of the polystomatic sublingual gland in captive and free-ranging Eurasian wolves.**
at–adipose tissue, c–capsule, ild–intralobular duct, is–interlobar septa, l–lobes, lym–lymphocytes, mu–mucous units, sd–striated duct, white asterisk–serous demilunes. a–f = H&E stain; e = mucicarmine stain. Bar = a– 200 μm; b– 100 μm; d, e– 50 μm; c, f– 20 μm.

the domestic dog [23], on the first upper molar tooth in the maned wolf [29] and fennec fox [30]. In the South African painted dog, it opened at the height of the first and second upper molar [30]. According to Souza et al. [28], in the crab-eating fox and Pampas fox, the parotid ducts opened between the third and fourth upper premolars in 53.6% of cases and at the level of the first upper molar in 46.4% of cases. These authors suggested that variations in the location of the parotid papilla might be due to rigor mortis, formalin fixation, or the method of restraining living animals, as these factors can affect the position of the papilla during examination.

The microscopic study in free-ranging and captive wolves showed that the parotid gland was a complex branched alveolar gland with serous secretion, similar to the domestic dog [10,12,21], Baladi dog [26] and fennec fox [30] while in the red fox [26] and in the South African painted dog [30] this gland was characterized by serous and mucous secretory units. The duct system of the parotid gland was well-developed in all examined wolves, similarly to the aforementioned species. Moreover, in female captive wolves, the presence of numerous lymphocytes in the parotid gland was found, which may indicate an ongoing inflammatory process in this salivary gland.

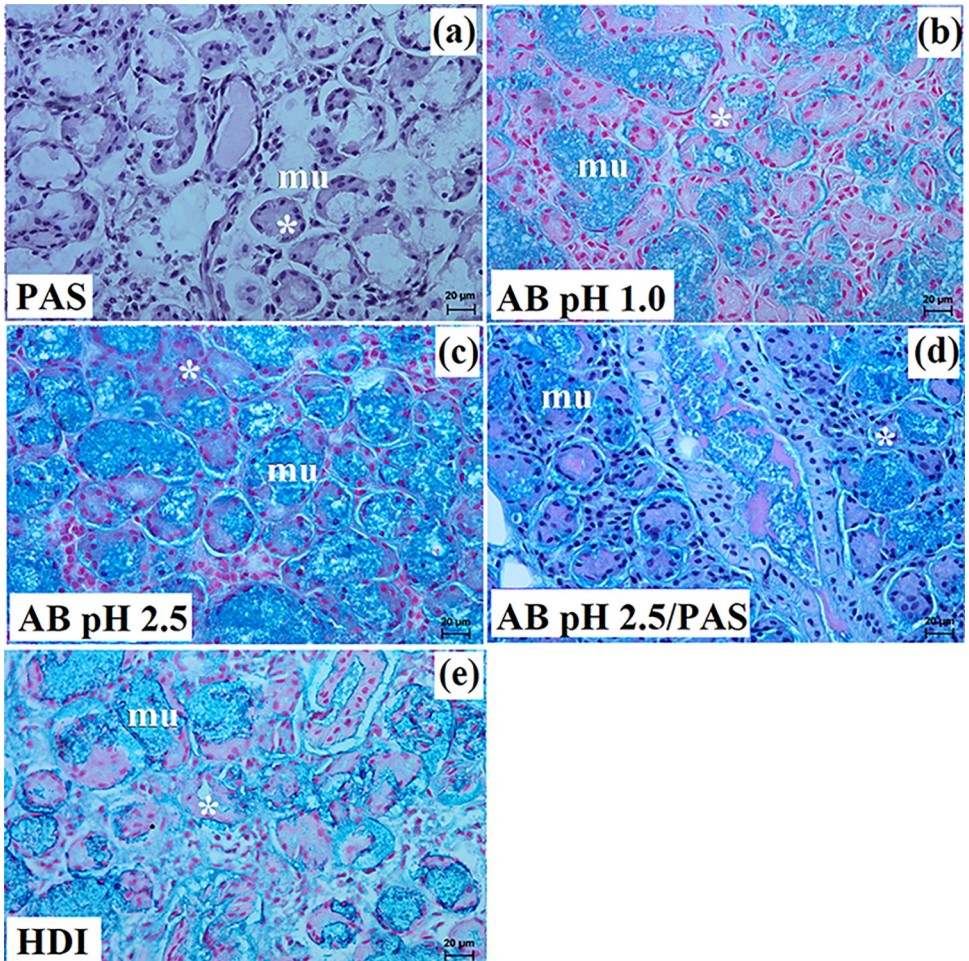

**Fig 11. The histochemical images of the polystomatic sublingual gland in captive and free-ranging Eurasian wolves.** mu–mucous units, white asterisk–serous demilunes. Bar = 20 μm.

## The mandibular gland

The mandibular gland in both captive and free-ranging wolves was situated in the retromandibular fossa, similar to a domestic dog [10,11,23,24], Baladi dog [26], crab-eating fox [27], red fox [26] and South African painted dog [30]. Whereas in the fennec fox, it was located behind the ramus of the mandible in the ventral neck region on the surface of the sternocleidomastoid muscle [30]. In our wolves, the dorsal margin of the mandibular gland was partially covered by the parotid gland, similar to the crab-eating fox [27].

Both in captive and free-ranging wolves, the mandibular gland was larger than the parotid gland, consistent with the description in the domestic dog as reported by Krysiak and Świeżyński [51]. Nazih and El-Sherif [16] report that in the Baladi dog, and similar to Gaber et al. [10] and Tadjalli et al. [23] in mongrel dogs, and Klećkowska-Nawrot et al. [30] in South African painted dog and fennec fox, the dimensions of the mandibular gland were notably smaller than in examined wolves.

In our wolves, this gland had an oval shape with a slightly concave rostral margin similar to the domestic dog [10,24]. The spherical shape was observed in the crab-eating fox [27]. However, Nazih and El-Sherif [16] and Klećkowska-Nawrot et al. [30] report, that in the Baladi dog

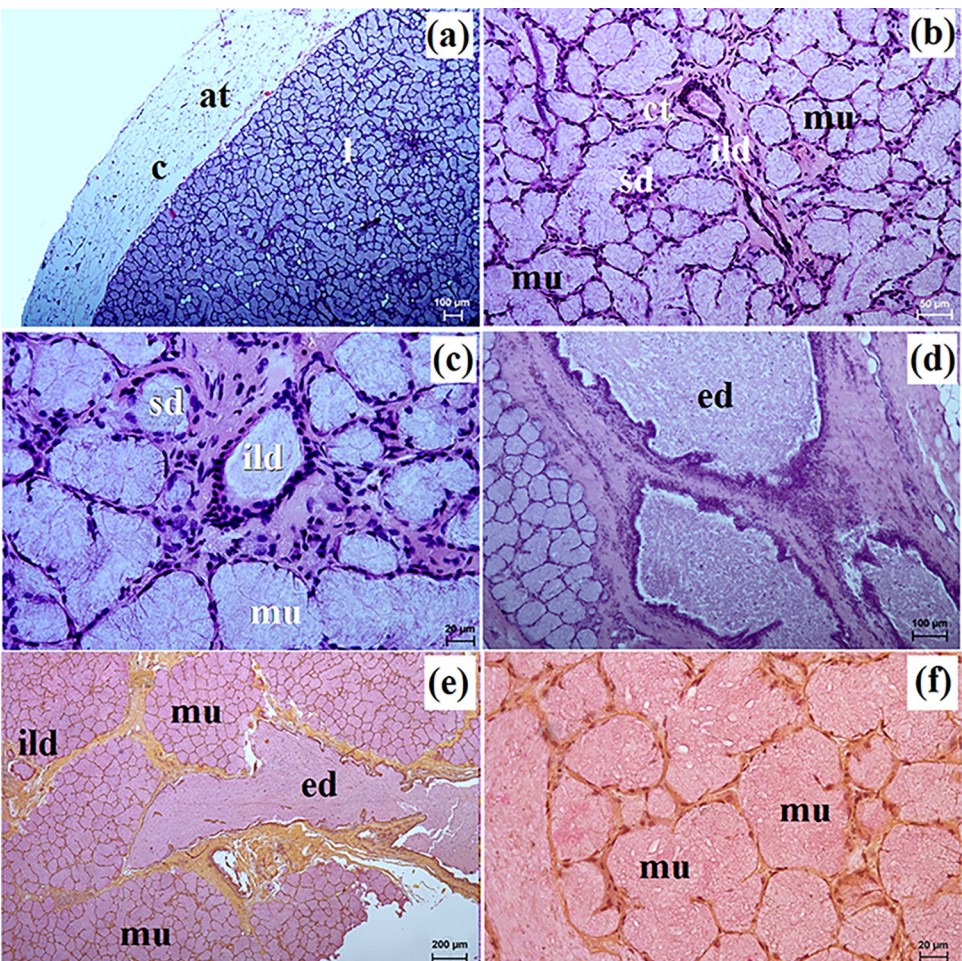

**Fig 12. The histological images of the zygomatic gland in free-ranging Eurasian wolves.** at–adipose tissue, c–capsule, ct–connective tissue, ed–excretory duct, ild–intralobular duct, is–interlobar septa, l–lobes, mu–mucous units, sd–striated duct. a–d = H&E stain; e, f = mucicarmine stain. Bar = a– 100 µm; b– 50 µm; c, f– 20 µm; e– 200 µm.

and South African painted dog, the mandibular gland has an oval to elliptical outline. The spherical shape was also observed crab-eating fox [27] and fennec fox [30]. The mandibular duct in wolves had a similar course to the domestic dog, crab-eating fox, fennec fox and South African painted dog [2,3,23,27,29,30].

The histological and histochemical studies showed that the mandibular gland in captive wolves was a branched complex tubular gland producing a mucous secretion, similarly to South African painted dog [30]. However, in free-ranging wolves, there was a branched tubuloalveolar system producing mucoserous secretions, similar to domestic dog and fennec fox [10,12,18,21,22,30,53]. Moreover, in 1 male and 3 female free-ranging wolves, the presence of numerous lymphocytes located around the intralobular ducts was observed, which may indicate an ongoing inflammatory process in this gland.

## The monostomatic sublingual gland

The monostomatic sublingual gland in our wolves was located in the lateral sublingual recess of the oral cavity proper, similar to the domestic dog [10,11,24], crab-eating fox [27], fennec fox and South African painted dog [30].

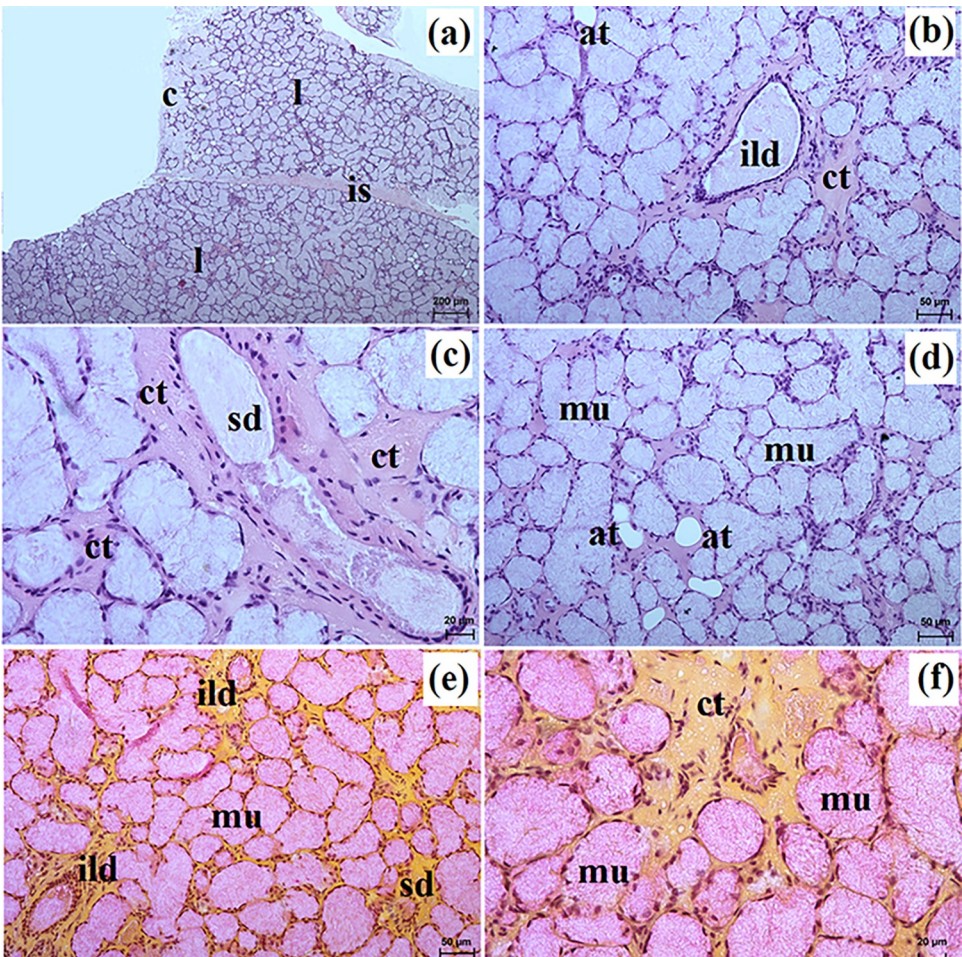

**Fig 13. The histological images of the zygomatic gland in captive Eurasian wolves.** at–adipose tissue, c–capsule, ct–connective tissue, ild–intralobular duct, is–interlobar septa, l–lobes, mu–mucous units, sd–striated duct. a–d = H&E stain; e, f = mucicarmine stain. Bar = a– 200 μm; b, d, e– 50 μm; c, f– 20 μm.

The size of the monostomatic sublingual gland in the examined captive and free-ranging wolves was very similar. According to Gaber et al. [10], in mongrel dogs, this gland was smaller than in our wolves, which may be due to the different body sizes of the dogs tested.

The shape of this gland in captive and free-ranging wolves was oval with a slightly marked indentation on the rostral edge, similar to that of breed and mongrel dogs [24], while in mongrel dogs examined by Gaber et al. [10], the outline of the major sublingual gland resembled an elongated triangle. However, the shape of this gland in fennec fox and South African painted dog was elongated [30].

In the wolves studied, similarly to the domestic dog, crab-eating fox, fennec fox and South African painted dog the major sublingual duct ran together with the mandibular duct and entered the oral cavity proper at the sublingual caruncle [10,27,30]. Our study showed that in male free-ranging wolves, the major sublingual duct was longer than in both captive and free-ranging female wolves studied. Histological and histochemical studies showed that in free-ranging wolves, this sublingual gland was a branched tubuloalveolar complex gland producing seromucous secretion, with a clear predominance of serous cells similar to the domestic dog [12] and fennec fox [30]. Gaber et al. [10] reports that the domestic dog also had mixed

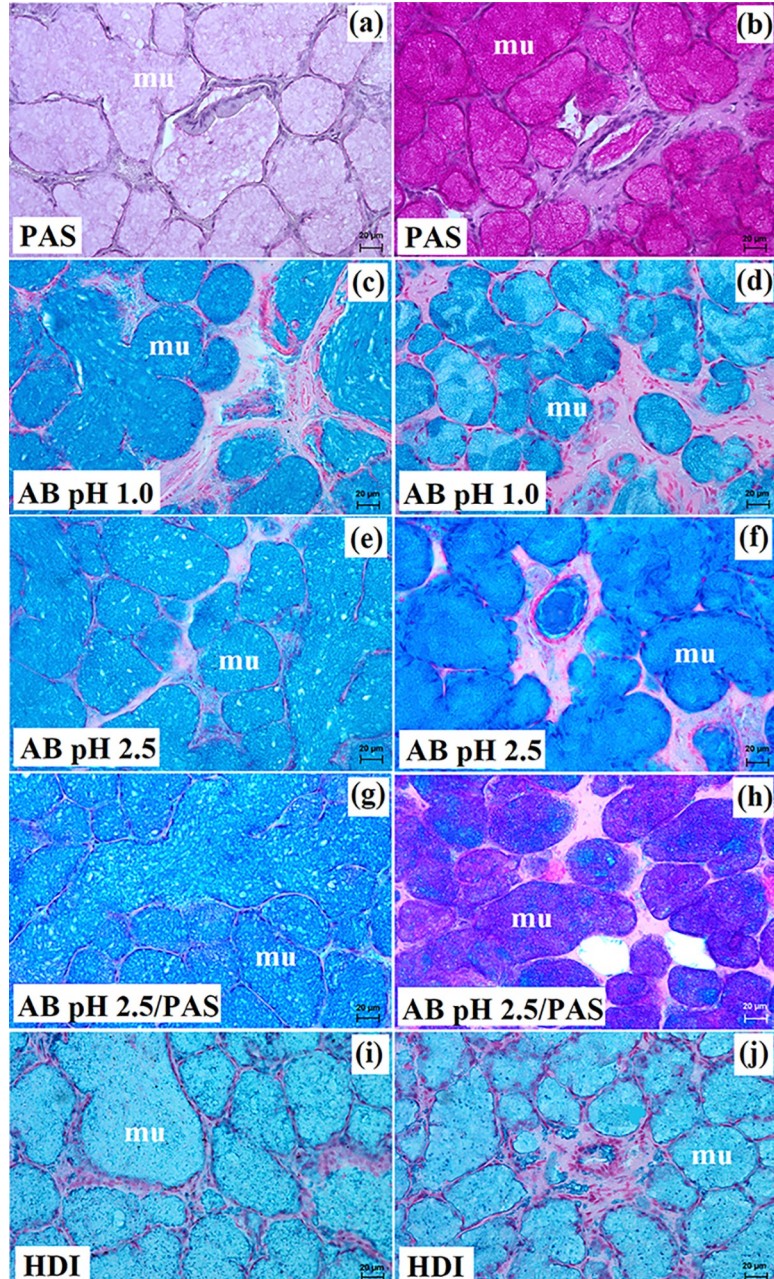

**Fig 14.** The histochemical images of the zygomatic gland in free-ranging Eurasian wolves (a, c, e, g, i) and captive wolves (b, d, f, h, j). mu–mucous units. Bar = 20 μm.

secretion as well as mucous acini and serous demilunes of nearly equal density. However, in captive wolves, it was a complex branched tubuloalveolar gland with a predominance of tubules and producing a mucoserous secretion. In the monostomatic sublingual gland of 1 male and 3 female free-ranging wolves, numerous lymphocytes were also observed. The striated ducts and intralobular ducts in wolves of the mandibular gland and monostomatic lingual glands were also well-developed, similar to the domestic dog [12].

## The polystomatic sublingual gland

The localization of this oral cavity gland in all examined wolves was similar to that observed in the domestic dog, the crab-eating fox, the fennec fox and the South African painted dog [3,10,27,30].

In our wolves, this salivary gland was narrow and long, and morphometric measurements showed that the size of the gland was very similar in all individuals. It consisted of several independent packets (6–7 to 7–8), each of which had its own excretory duct leading directly to the floor of the oral cavity proper.

Histological and histochemical analysis showed that in all examined wolves, it was a complex branched tubuloalveolar gland producing seromucous secretion similar to the domestic dog [21]. In captive wolves, numerous clusters of lymphocytes were visible around striated ducts and secretory units.

## The zygomatic gland

The topography of this gland in studied wolves corresponded to its location in the domestic dog and South African painted dog [3,10,11,23,30].

Morphometric measurements among the examined individuals showed that the zygomatic gland was longer in free-ranging male wolves than in all females, while in terms of width and thickness, it was similar in all wolves. These results are comparable to those observed in the domestic dog [10].

In our wolves, the zygomatic gland was oval in shape, while in the domestic dog (breed dogs, mesocephalic dogs, and mongrel dogs), it was conical in shape [10,24,54,55]. In the fennec fox, the South African painted dog and the Philippine non-descript dog, it was irregular in shape [30,56].

The zygomatic gland duct in our wolves had 4 to 5 secretory ducts that opened at the height of the last upper molar, similar to the domestic dog [23].

The zygomatic gland in examined wolves was a branched tubular complex gland producing a mucous-like secretion similar to the domestic dog [10,19,21,22,56], the fennec fox and the South African painted dog [30]. According to Gomi et al. [12] and Mohamed [55], in the domestic dog, this secretion was also mixed.

## Conclusions

The aforementioned studies have shown that the general anatomical and histological structure, including histochemical analysis of the major salivary glands and zygomatic gland in both free-ranging and captive wolves is comparable to that described in other representatives of terrestrial carnivores. Our research has indicated that the degree of similarity in the position, shape, histological and histochemical structure of the major salivary glands and zygomatic gland between captive wolves, free-ranging wolves and other terrestrial carnivores may likely result from their phylogenetic origin. According to Ackerly and Donoghue [57], these animals are probably very similar in their phylogeny, suggesting that they may come from the same common ancestor. Despite the diverse diet and environment in which they live, this indicates a high degree of conservation of anatomical features found in these species, supporting the hypothesis of common origin, where current forms derive from a possible single common ancestor.

In addition, these results provide information on similarities and morphological differences in oral cavity glands between free-ranging and captive wolf populations resulting from the influence of the environment in which they live and, therefore, access to a diversified diet. Another aspect of these studies is also a helpful tool for future veterinary treatments involving

salivary glands, as well as a reference point for determining pathological conditions related to the major salivary glands and zygomatic glands themselves, including their main output ducts, but also pathological changes occurring within them.

These studies also indicate the advisability of conducting more research on individuals representing other wolf populations, whose diet contains a different species composition closely related to the environment in which a given wolf population occurs. To best understand the diversity in the structure of salivary glands, we should also take into account individual subspecies of *Canis lupus*, including the domestic dog and individual breeds (different diet and body size). Moreover, such research should also be extended to include other representatives of the *Canis* genus and other species occurring within the Canidae family. The *Canis* genus has a very wide range, so the analysis of individual populations would allow us to better understand the influence of diet on the morphology of salivary glands.

## Supporting information

**S1 Table. Measurements (mm) of the salivary glands of the Eurasian wolf.** The length, width and thickness of the mandibular, parotid, zygomatic and monostomatic sublingual glands as well as their ducts length.
(XLSX)

## Acknowledgments

We extend our gratitude to the association "Z Szarym za Płotem", Mr. Paweł Kowalczyk and Mr. Jan Fiderewicz for the help with obtaining study materials. Additionally, we appreciate the assistance of DVM Wojciech Paszta, Ph.D. and DVM Krzysztof Zagórski from the Wroclaw Zoological Garden for providing valuable study materials. Furthermore, we would like to thank Mr. Tomasz Filipiak and Mr. Piotr Krawczyk, senior specialists in the Division of Animal Anatomy, Department of Biostructure and Animal Physiology, for their help in capturing anatomical photographs.

## Author Contributions

**Conceptualization:** Joanna Klećkowska-Nawrot, Karolina Goździewska-Harłajczuk.

**Data curation:** Joanna Klećkowska-Nawrot, Arkadiusz Dziech.

**Formal analysis:** Joanna Klećkowska-Nawrot, Karolina Goździewska-Harłajczuk, Krzysztof Stegmann, Dariusz Łupicki.

**Funding acquisition:** Karolina Barszcz.

**Investigation:** Joanna Klećkowska-Nawrot, Arkadiusz Dziech, Gabriela Jędrszczyk, Igor Jucenco.

**Methodology:** Joanna Klećkowska-Nawrot, Krzysztof Stegmann, Arkadiusz Dziech, Dariusz Łupicki, Gabriela Jędrszczyk, Igor Jucenco.

**Project administration:** Karolina Barszcz.

**Resources:** Dariusz Łupicki.

**Visualization:** Joanna Klećkowska-Nawrot, Karolina Goździewska-Harłajczuk, Gabriela Jędrszczyk, Igor Jucenco, Karolina Barszcz.

**Writing – original draft:** Joanna Klećkowska-Nawrot, Karolina Goździewska-Harłajczuk, Krzysztof Stegmann, Arkadiusz Dziech.

**Writing – review & editing:** Karolina Goździewska-Harłajczuk.

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
