## [Decision Letter · Decision Letter 0]

27 Sep 2024

PONE-D-24-28995Anatomical Variations in Eurasian Wolf (Canis lupus lupus) (Carnivora: Canidae) of the Salivary Glands: A Histological and Histochemical InvestigationPLOS ONE

Dear Dr. Goździewska-Harłajczuk,

Thank you for submitting your manuscript to PLOS ONE. After careful consideration, we feel that it has merit but does not fully meet PLOS ONE’s publication criteria as it currently stands. Therefore, we invite you to submit a revised version of the manuscript that addresses the points raised during the review process.

We look forward to receiving your revised manuscript.

Kind regards,

Anindita Bhadra, PhD

Academic Editor

PLOS ONE

Journal Requirements: When submitting your revision, we need you to address these additional requirements. 1. Please ensure that your manuscript meets PLOS ONE's style requirements, including those for file naming. The PLOS ONE style templates can be found at https://journals.plos.org/plosone/s/file?id=wjVg/PLOSOne_formatting_sample_main_body.pdf and https://journals.plos.org/plosone/s/file?id=ba62/PLOSOne_formatting_sample_title_authors_affiliations.pdf 2. We note that the grant information you provided in the ‘Funding Information’ and ‘Financial Disclosure’ sections do not match.  When you resubmit, please ensure that you provide the correct grant numbers for the awards you received for your study in the ‘Funding Information’ section. 3. Thank you for stating the following financial disclosure: "The APC was financed by Warsaw University of Life Sciences"  Please state what role the funders took in the study.  If the funders had no role, please state: ""The funders had no role in study design, data collection and analysis, decision to publish, or preparation of the manuscript."" If this statement is not correct you must amend it as needed. Please include this amended Role of Funder statement in your cover letter; we will change the online submission form on your behalf. 4. We note that your Data Availability Statement is currently as follows: All relevant data are within the manuscript and its Supporting Information files. Please confirm at this time whether or not your submission contains all raw data required to replicate the results of your study. Authors must share the “minimal data set” for their submission. PLOS defines the minimal data set to consist of the data required to replicate all study findings reported in the article, as well as related metadata and methods (https://journals.plos.org/plosone/s/data-availability#loc-minimal-data-set-definition). For example, authors should submit the following data: - The values behind the means, standard deviations and other measures reported;- The values used to build graphs;- The points extracted from images for analysis. Authors do not need to submit their entire data set if only a portion of the data was used in the reported study. If your submission does not contain these data, please either upload them as Supporting Information files or deposit them to a stable, public repository and provide us with the relevant URLs, DOIs, or accession numbers. For a list of recommended repositories, please see https://journals.plos.org/plosone/s/recommended-repositories. If there are ethical or legal restrictions on sharing a de-identified data set, please explain them in detail (e.g., data contain potentially sensitive information, data are owned by a third-party organization, etc.) and who has imposed them (e.g., an ethics committee). Please also provide contact information for a data access committee, ethics committee, or other institutional body to which data requests may be sent. If data are owned by a third party, please indicate how others may request data access. 5. Please include your tables as part of your main manuscript and remove the individual files. Please note that supplementary tables (should remain/ be uploaded) as separate ""supporting information"" files 6. Please review your reference list to ensure that it is complete and correct. If you have cited papers that have been retracted, please include the rationale for doing so in the manuscript text, or remove these references and replace them with relevant current references. Any changes to the reference list should be mentioned in the rebuttal letter that accompanies your revised manuscript. If you need to cite a retracted article, indicate the article’s retracted status in the References list and also include a citation and full reference for the retraction notice. 

**Additional Editor Comments:**

Your manuscript has now been reviewed by two independent reviewers. As you would see that only one of them has suggested minor revision. I would like to request you to address these comments and submit a revised version of your manuscript for consideration of publication.

Reviewers' comments:

Reviewer's Responses to Questions

**Comments to the Author**

1. Is the manuscript technically sound, and do the data support the conclusions?

Reviewer #1: Yes

Reviewer #2: Yes

2. Has the statistical analysis been performed appropriately and rigorously? 

Reviewer #1: Yes

Reviewer #2: N/A

3. Have the authors made all data underlying the findings in their manuscript fully available?

Reviewer #1: Yes

Reviewer #2: Yes

4. Is the manuscript presented in an intelligible fashion and written in standard English?

Reviewer #1: Yes

Reviewer #2: Yes

5. Review Comments to the Author

Reviewer #1: The work provides important data on the morphology of the salivary glands of Eurasian Wolf. The methodology used is appropriate for the proposed objectives and the conclusion is concise with the results obtained, which were described in detail. I would like to highlight the quality of the anatomical images and the photomicrographs.

Reviewer #2: The work entitled “Anatomical Variations in Eurasian Wolf (Canis lupus lupus) (Carnivora Canidae) of the Salivary Glands A Histological and Histochemical Investigation” with ID: PONE-D-24-28995 has been designed and conducted well, with a remarkable conclusion. However, the authors should address some issues before considering the manuscript for publication (minor revision). The manuscript should be revised for linguistic, grammatical, word spacing, and style errors. The format of the whole manuscript and the reference style must comply with the guidelines of the Journal.

Title:

Good

Abstract:

good

Introduction:

Very good

Material and method:

Good.

Results:

good However the quality of figures should be increased as much as possible

Discussion:

good

Good luck

6. PLOS authors have the option to publish the peer review history of their article (what does this mean?). If published, this will include your full peer review and any attached files.

Reviewer #1: No

Reviewer #2: No

---

## [Author Response · Author response to Decision Letter 0]

25 Oct 2024

Response to Editor and Reviewers

We would like to thank the Editor and Reviewers for the comments on the paper, as these comments led us to an improvement of the work. The manuscript has been corrected with the recommendations of Editor and Reviewer 2. I`m sending to You answers for these comments. All the changes were signed in red. 

Answer for the Comments of Editor:

Comment 1. “Please ensure that your manuscript meets PLOS ONE's style requirements, including those for file naming. The PLOS ONE style templates can be found at 

https://journals.plos.org/plosone/s/file?id=ba62/PLOSOne_formatting_sample_title_authors_affiliations.pdf”

Answer: The style of our manuscript was improved and the current version meet the PLOS ONE`s style.

Comment 2. “We note that the grant information you provided in the ‘Funding Information’ and ‘Financial Disclosure’ sections do not match. When you resubmit, please ensure that you provide the correct grant numbers for the awards you received for your study in the ‘Funding Information’ section.”

Answer: The Funding information was corrected and unified.

Comment 3. “Thank you for stating the following financial disclosure: "The APC was financed by Warsaw University of Life Sciences. "Please state what role the funders took in the study. If the funders had no role, please state: ""The funders had no role in study design, data collection and analysis, decision to publish, or preparation of the manuscript."" If this statement is not correct you must amend it as needed. Please include this amended Role of Funder statement in your cover letter; we will change the online submission form on your behalf.”

Answer: The sentence: “The funders had no role in study design, data collection and analysis, decision to publish, or preparation of the manuscript” was added in the text of the manuscript.

Comment 4. “We note that your Data Availability Statement is currently as follows: All relevant data are within the manuscript and its Supporting Information files. Please confirm at this time whether or not your submission contains all raw data required to replicate the results of your study. Authors must share the “minimal data set” for their submission. PLOS defines the minimal data set to consist of the data required to replicate all study findings reported in the article, as well as related metadata and methods (https://journals.plos.org/plosone/s/data-availability#loc-minimal-data-set-definition).”

 Answer: The manuscript contain all relevant data. We added the supplementary data as Supplementary file.

Comment 5. “Please include your tables as part of your main manuscript and remove the individual files. Please note that supplementary tables (should remain/ be uploaded) as separate ""supporting information"" files.”

 Answer: The tables were included in the text. The individual files were removed.

Comment 6. “Please review your reference list to ensure that it is complete and correct. If you have cited papers that have been retracted, please include the rationale for doing so in the manuscript text, or remove these references and replace them with relevant current references. Any changes to the reference list should be mentioned in the rebuttal letter that accompanies your revised manuscript. If you need to cite a retracted article, indicate the article’s retracted status in the References list and also include a citation and full reference for the retraction notice.”

Answer: The Reference list was corrected.

Answer for Reviewers:

Reviewer 1

Reviewer #1: “The work provides important data on the morphology of the salivary glands of Eurasian Wolf. The methodology used is appropriate for the proposed objectives and the conclusion is concise with the results obtained, which were described in detail. I would like to highlight the quality of the anatomical images and the photomicrographs.”

Answer: Dear Reviewer thank you very much for the opinion on the our manuscript.

Reviewer 2

Reviewer #2: “The work entitled “Anatomical Variations in Eurasian Wolf (Canis lupus lupus) (Carnivora Canidae) of the Salivary Glands A Histological and Histochemical Investigation” with ID: PONE-D-24-28995 has been designed and conducted well, with a remarkable conclusion. However, the authors should address some issues before considering the manuscript for publication (minor revision). The manuscript should be revised for linguistic, grammatical, word spacing, and style errors. The format of the whole manuscript and the reference style must comply with the guidelines of the Journal. Title: good, Abstract: good, Introduction: very good, Material and method: very good.

Results: good However the quality of figures should be increased as much as possible. Macroscopic images were recaptured., Discussion: good.”

Answer: Dear Reviewer thank you for the opinion on the our manuscript. The text was revised for linguistic, grammatical, word spacing and style errors. The format of the manuscript complies with the guidelines of the Journal, what was corrected. The quality of all figures was improved. All the changes were marked in the text in red.

We are looking forward to your opinion on the paper.

Sincerely Yours,

Karolina Goździewska-Harłajczuk

---

## [Decision Letter · Decision Letter 1]

22 Dec 2024

Anatomical variations in Eurasian wolf (Canis lupus lupus) (Carnivora: Canidae) of the salivary glands: A histological and histochemical investigation

PONE-D-24-28995R1

Dear Dr. Goździewska-Harłajczuk,

We’re pleased to inform you that your manuscript has been judged scientifically suitable for publication and will be formally accepted for publication once it meets all outstanding technical requirements.

Kind regards,

Serkan Erdoğan, Ph.D.

Academic Editor

PLOS ONE

Additional Editor Comments (optional):

Reviewers' comments:

Reviewer's Responses to Questions

**Comments to the Author**

1. If the authors have adequately addressed your comments raised in a previous round of review and you feel that this manuscript is now acceptable for publication, you may indicate that here to bypass the “Comments to the Author” section, enter your conflict of interest statement in the “Confidential to Editor” section, and submit your "Accept" recommendation.

Reviewer #1: All comments have been addressed

Reviewer #2: (No Response)

2. Is the manuscript technically sound, and do the data support the conclusions?

Reviewer #1: Yes

Reviewer #2: Yes

3. Has the statistical analysis been performed appropriately and rigorously? 

Reviewer #1: Yes

Reviewer #2: Yes

4. Have the authors made all data underlying the findings in their manuscript fully available?

Reviewer #1: Yes

Reviewer #2: Yes

5. Is the manuscript presented in an intelligible fashion and written in standard English?

Reviewer #1: Yes

Reviewer #2: Yes

6. Review Comments to the Author

Reviewer #1: The work is great and that's why I recommend your publication. Congratulations to the authors for the quality of the research developed.

Reviewer #2: The required modifications for the work entitled "Anatomical variations in Eurasian wolf (Canis lupus lupus) (Carnivora: Canidae) of the salivary glands: A histological and histochemical investigation, was done. So, we recommend the acceptance in the current form.

Greetings

7. PLOS authors have the option to publish the peer review history of their article (what does this mean?). If published, this will include your full peer review and any attached files.

Reviewer #1: No

Reviewer #2: No

---

## [Editor Report · Acceptance letter]

28 Dec 2024

PONE-D-24-28995R1 

PLOS ONE

Dear Dr. Goździewska-Harłajczuk, 

I'm pleased to inform you that your manuscript has been deemed suitable for publication in PLOS ONE. Congratulations! Your manuscript is now being handed over to our production team.

Kind regards, 

on behalf of

Dr. Serkan Erdoğan 

Academic Editor

PLOS ONE